# Hierarchical Multimodal Knowledge Matching for Training-Free Open-Vocabulary Object Detection

## Abstract

Open-Vocabulary Object Detection (OVOD) aims to leverage the generalization capabilities of pre-trained vision language models for detecting objects beyond the trained categories. Existing methods mostly focus on supervised learning strategies based on available training data, which might be suboptimal for data-limited novel categories. To tackle this challenge, this paper presents a **H**ierarchical **M**ultimodal **K**nowledge **M**atching method (**HMKM**) to better represent novel categories and match them with region features. Specifically, HMKM includes a set of object prototype knowledge that is obtained using limited category-specific images, acting as off-the-shelf category representations. In addition, HMKM also includes a set of attribute prototype knowledge to represent key attributes of categories at a fine-grained level, with the goal to distinguish one category from its visually similar ones. During inference, two sets of object and attribute prototype knowledge are adaptively combined to match categories with region features. The proposed HMKM is training-free and can be easily integrated as a plug-and-play module into existing OVOD models. Extensive experiments demonstrate that our HMKM significantly improves the performance when detecting novel categories across various backbones and datasets.

## 1 Introduction

Object detection is a core computer vision task that involves localizing and classifying objects in images (Ren et al., 2015; Lin et al., 2017; He et al., 2017; Cai & Vasconcelos, 2018). Traditional methods are limited to predefined categories, which makes them less practical in real-world setting. Open-Vocabulary Object Detection (OVOD) models (Kamath et al., 2021; Li et al., 2022; Cai et al., 2022) expand the range of detectable categories using pre-trained Vision Language Models (VLMs) to align visual region features with textual category features.

Existing OVOD methods are mostly training-based, focusing on knowledge distillation from VLMs (Gu et al., 2022; Wang et al., 2023b; Wu et al., 2023a), incorporating learned prompts into classifiers (Du et al., 2022; Feng et al., 2022; Wu et al., 2023b), improving region-text alignment (Zhong et al., 2022; Lin et al., 2023; Ma et al., 2024a), and generating detailed textual descriptions of categories (Kaul et al., 2023; Jin et al., 2024; Kim et al., 2024). The key to their success is the supervised learning on large-scale image-text pair datasets, which can better match region features and textual category features. However, these methods might have the following limitations, as shown in Figure 1a. 1) They struggle to learn effective representations for novel categories, resulting in lower performance compared to the average performance on base categories, due to the limited number of pairwise samples. 2) Even with additionally generated textual descriptions, detection of novel categories remains much lower than the average on base categories, as these descriptions fail to capture fine-grained visual details. How to deal with these problems is very important but rarely investigated.

In contrast to existing methods, human brains do not need such a large number of pair samples for supervised learning. They learn and comprehend novel concepts mainly using multimodal knowledge stored in the long-term memory, in which visual objects and attributes representations are associated with linguistic categories in a prototypical manner (Tulving, 1972; Bi, 2021). As shown

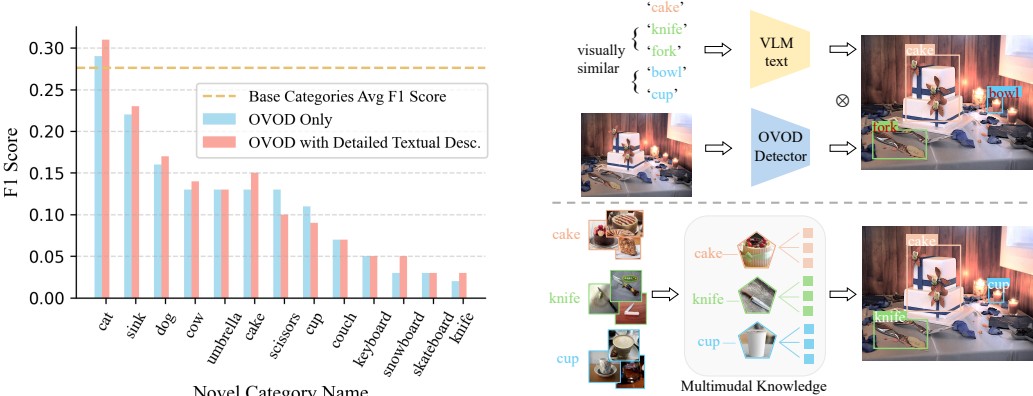

(a) F1 Scores for novel categories on COCO dataset.  (b) Illustration of OVOD and brain mechanisms.

Figure 1: Limitations of OVOD in detecting novel categories and and a comparative illustration with the learning mechanisms of the human brains. (a) Presents the F1 score statistics across different settings for novel categories, based on the recent OVOD method VLDet (Lin et al., 2023). (b) Compares the mechanisms of OVOD and human brains for learning novel categories.

in Figure 1b, human brains can use multimodal knowledge to represent novel categories from limited images, and accurately detect them off-the-shelf. Although there are works (Wang et al., 2020a; Zhang et al., 2021; Ding et al., 2022) attempt to model the knowledge for other tasks, their knowledge modeling has the following limitations that are not suitable for OVOD. 1) Their knowledge can only be used during the training process of other supervised learning models, which is unsuitable for data-limited novel categories. 2) Their knowledge aligns each category to multiple object samples in a one-to-many manner, which could lead to confusion when dealing with objects with similar appearances.

To deal with the issues, this work proposes a **H**ierarchical **M**ultimodal **K**nowledge **M**atching method (**HMKM**), which can be used as an off-the-shelf module for representing novel categories and then matching them with region features. Initially, it selects a few images per category to build a object prototype knowledge set acting as off-the-shelf category representations for object-level matching. To further distinguish categories with similar appearances, the HMKM additionally creates an attribute prototype knowledge set by randomly cropping category images and clustering their attributes. In this way, attribute-level matching is performed to uncover fine-grained visual details. In summary, the proposed HMKM is a hierarchical matching strategy: object-level and attribute-level. This hierarchical multimodal knowledge can effectively represent categories and supplement textual descriptions. By combining the matching scores from two-level matchings, the detection capability for novel categories could be improved. Our HMKM as a plug-and-play module allows training-free integration into existing OVOD models during inference, which can consistently improve their performance. We validate our HMKM on the COCO and LVIS datasets, extensive experiments clearly demonstrate its effectiveness.

Our contributions are summarized as follows. 1) We develop a hierarchical multimodal knowledge matching method named HMKM, which can not only detect novel categories in a training-free manner, but also be easily integrated into existing OVOD models for further performance improvements. 2) We propose using object prototype knowledge for object-level feature alignment and attribute prototype knowledge for fine-grained matching. 3) Extensive experiments on the COCO and LVIS datasets demonstrate that our method consistently improves the performance when detecting novel categories across various backbones and datasets.

## 2 RELATED WORKS

**Few-Shot Object Detection.** Few-shot object detection aims to enhance a model's detection capabilities using only a few samples with annotated bounding box. Various methods have been proposed

to push forward research in this direction (Li et al., 2021; Lee et al., 2022; Wu et al., 2021). Current methods (Wang et al., 2020b; Qiao et al., 2021; Sun et al., 2021; Kaul et al., 2022) mainly improve detection performance of few-shot categories by fine-tuning the detector's parameters. These methods differ from our work focusing on the open-vocabulary object detection, where samples with bounding box annotations for novel categories are not used to update the model's parameters during training.

**Open-Vocabulary Object Detection.** Open-vocabulary object detection leverages the generalization capabilities of pre-trained VLMs to enhance object detection, allowing it to identify a wide range of novel categories and reduce laborious human annotations. Recent research in OVOD focuses on several directions as follows. Knowledge distillation methods (Gu et al., 2022; Wang et al., 2023b; Wu et al., 2023a) align region features with VLMs-derived features, seeking to transfer VLMs' multimodal representation capabilities to the model. Prompting modeling approaches (Du et al., 2022; Feng et al., 2022; Jin et al., 2024; Kaul et al., 2023) refine the textual embedding space of VLMs to better match with region features, incorporating richer prompts to transfer its knowledge to downstream tasks more easily. Region-text alignment methods (Zareian et al., 2021; Zhong et al., 2022; Li et al., 2022; Lin et al., 2023; Ma et al., 2024a; Wang et al., 2023a) use large-scale image-text datasets under weak supervision to expand their detection vocabulary. Unlike the above training-based methods, our research aims to leverage hierarchical multimodal knowledge to improve the ability to detect novel categories without extra training.

**Multimodal Knowledge.** Currently, several studies explore to model multimodal knowledge for various vision and language understanding tasks. Ding et al. (Ding et al., 2022) retrieve related multimodal knowledge from existing knowledge graphs, effectively linking visual objects with factual answers in the task of fact-based visual question answering. Wang et al. (Wang et al., 2020a) extract useful knowledge from multimodal data to identify discriminative parts of objects in the task of few-shot learning. Zhang et al. (Zhang et al., 2021) propose a concept-relation graph, composed of recursively combined semantic concepts, for the task of visual grounding. Different from above using image-word multimodal knowledge, Huang et al. further introduces more accurate region-word multimodal knowledge to improve image-text matching (Huang et al., 2022). Unlike these methods, we employ multimodal knowledge to a different task as OVOD. What's more, the multimodal knowledge we constructed is hierarchical including both object-level and attribute-level and could be easily integrated into the existing OVOD models in a training-free manner.

## 3 METHOD

In this section, we will explain the proposed HMKM for the task of OVOD. Before we dive into HMKM, we briefly introduce the OVOD task in Section 3.1.

The overall pipeline of the proposed HMKM is illustrated in Figure 2. HMKM comprises two-level matchings: 1) object prototype matching (OPM), which acts as off-the-shelf category representations and can match them with region features, and 2) attribute prototype matching (APM), which represents key attributes of categories at a fine-grained level and enhances the matching of visually similar categories. During inference, the proposed HMKM can also be used to improve the performance of OVOD models in a plug-and-play manner. The details of matchings are presented in Section 3.2 and Section 3.3, respectively.

### 3.1 PRELIMINARIES

Given an image $I \in \mathbb{R}^{3 \times h \times w}$, object detection aims to locate objects, represent each with bounding-box coordinates $b_j \in \mathbb{R}^4$ and assign a class label $c_j \in C^{test}$. Traditional models train and test on the same category set, i.e., $C^{test} = C^{base}$, while open-vocabulary object detection expands the test set to include both base and novel categories, i.e., $C^{test} = C^{base} \cup C^{novel}$. Most recent open-vocabulary object detectors use a two-stage architecture. Initially, a learned region proposal network (RPN) is used to generate $M$ region proposals $\{z_m\}_{m=1}^M = \Phi_{RPN}(I)$ from an image $I$, where each $z_m \in \mathbb{R}^D$ is a $D$-dimensional region-of-interest (RoI) feature embedding. Subsequently, a bounding box regressor predicts location coordinates for each region as $\hat{b}_m = \Phi_{REG}(z_m)$. Finally, the open-vocabulary classifier $\Phi_{CLS}(\cdot)$ computes classification scores using the cosine similarity, denoted as $s_m(c, z_m) = \langle w_c, z_m \rangle$, where each $w_c$ is encoded by a VLM text encoder such as CLIP (Radford

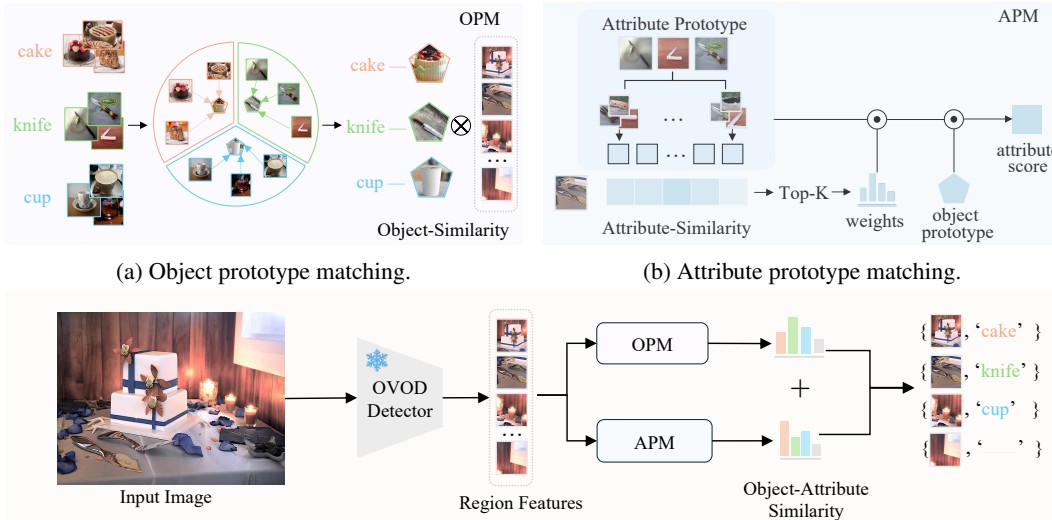

(a) Object prototype matching.  (b) Attribute prototype matching.

(c) Inference pipeline with HMKM.

Figure 2: Overall pipeline of HMKM. (a) **OPM**: It sequentially collects category names and their corresponding images, uses a frozen OVOD detector to extract image features, and obtains a object prototypical representation for each category by feature averaging. The object-similarity is computed through matrix multiplication between the category's prototype and the image's region features. (b) **APM**: It sequentially computes the similarity between each region feature and the category's attribute cluster features, selects the top-$k$ most similar attributes, weights these attributes by their similarity, and multiplies them by the category's object prototype to determine the attribute matching similarity score. (c) **Inference pipeline with HMKM.** First, inputing an image into the OVOD detector to extract multiple region features. Then, processing each region feature through OPM and APM to calculate object-similarity and attribute-similarity scores. Finally, combining these scores to assign the highest similarity category as the prediction for each region.

et al., 2021), representing class name embeddings in $C_{test}$. For the $m$-th region, the final predicted category $\hat{y}_m$ can be obtained using:

$$\hat{y}_m = \arg \max_{c \in C_{\text{test}}} \langle w_c, z_m \rangle \tag{1}$$

The overall open-vocabulary detection process can be formulated as follows:

$$\{\hat{y}_1, \ldots, \hat{y}_n\} = \Phi_{CLS} \circ \Phi_{REG} \circ \Phi_{RPN} \circ \Phi_{ENC}(I_i) \tag{2}$$

where $I_i$ denotes the $i$-th input image and $\{\hat{y}_1, \ldots, \hat{y}_n\}$ represents the set of predicted outputs. Our work primarily focuses on enhancing the classification process of the open-vocabulary classifier $\Phi_{CLS}(\cdot)$.

### 3.2 OBJECT PROTOTYPE MATCHING

As illustrated in Figure 2a, we sequentially introduce the collection of category word and image, the representation of category object prototype, and the matching of category object prototype.

**Category Word and Image Collection.** Assuming object detection test set contains $C^{test}$ categories, we align each category with the ImageNet-21k repository (Deng et al., 2009) using WordNet synsets (Miller, 1995). For each category in $C^{test}$, we randomly select $T$ images. Intuitively, common categories like "dog" and "cat" have many images to form effective object prototypes due to their diverse subtypes. Conversely, less common categories such as "banjo", which have fewer images, posing challenges to object prototype representation. However, our experiments show that even using limited images can still generate effective object prototypes.

**Category Object Prototype Representation.** For each category, semantically related objects in various regions usually have diverse visual appearances, potentially causing confusion. Instead

of linking each category to multiple related regions in a one-to-many manner, we represent each category as an object prototype to alleviate appearance variation issues. As illustrated in Figure 2a, for category $i$, we obtain its object prototype representation $p_i$ by averaging all related image features derived from the frozen OVOD backbone $\Phi_{det}$:

$$p_i = \frac{1}{T} \sum_{j=1}^{T} \Phi_{det}(I_j) \tag{3}$$

where $I_j$ refers to the $j$-th image of the category, and $T$ denotes the number of images. As a result, we obtain the paired object prototype knowledge $\{(c_i, p_i)\}_{i=1}^{S}$ where $c_i$ is the name of the $i$-th category in $C^{test}$, $p_i$ is the corresponding object prototype representation, and $S$ is the number of categories in $C^{test}$.

**Category Object Prototype Matching.** For the $m$-th region feature $z_m$ in an image $I$, the corresponding object prototype representation for each detected category is extracted from the object prototype knowledge set using its category name, and the object-level similarity for all categories is computed. The formula for category object prototype matching is as follows:

$$S_{\text{prot}}(p_c, z_m) = \langle p_c, z_m \rangle \tag{4}$$

where $\langle \cdot, \cdot \rangle$ denotes cosine similarity.

### 3.3 ATTRIBUTE PROTOTYPE MATCHING

The aforementioned object prototype matching primarily focuses on object-level matching, which might struggle to distinguish visually similar categories. To address this issue, we introduce attribute prototype matching to enhance the fine-grained matching. As illustrated in Figure 2b, we sequentially introduce the representation of category attribute prototype and the matching of category attribute prototype.

**Category Attribute Prototype Representation.** After collecting images for each category, we can further construct attribute prototype knowledge based on them. Unlike creating object prototypes, finding multiple informative attributes for each category, is more complex and requires more reference images. It is because a single image often fails to display all the necessary attributes of an object. Therefore, we increase the number of images $M$ used for generating attribute prototypes for each category. In particular, by performing $N$ random croppings on each image within a category, $N$ attribute regions are extracted per image. Consequently, for a category with $M$ images, a total of $M \times N$ attribute regions are obtained. Subsequently, using the visual backbone $\Phi_{det}$, features for these $M \times N$ attribute regions are extracted individually. Finally, by employing a clustering method and specifying the number of clusters $W$, cluster centers among the $W$ attribute region features are identified, serving as the categorical attribute prototypes. Note that we empirically demonstrate that the attribute prototypes obtained through clustering are more stable and effective than those constructed using the individual attribute regions directly. The attribute prototype representation $a_i$ for category $i$ is defined as follows:

$$a_i = \text{Cluster}\left(\{\Phi_{det}\left(\text{Crop}(I_{m,n})\right)\}_{m=1,n=1}^{M,N}\right) \tag{5}$$

where $a_i$ is a $2D$ vector consisting of $W$ separate $1D$ vectors for attribute prototype representations, denoted by $\{a_{i1}, a_{i2}, \ldots, a_{iW}\}$. The hierarchical multimodal knowledge, including attribute prototype knowledge, can further be formulated as a set of triples: $\{(c_i, p_i, a_i)\}_{i=1}^{S}$ where $c_i$ is the name of the $i$-th category in $C^{test}$, $p_i$ is the corresponding object prototype representation, and $S$ is the number of categories in $C^{test}$.

**Category Attribute Prototype Matching.** Due to each category containing multiple attribute prototypes with different importances, it is challenging to achieve matching results directly using simple cosine similarity. To address this, we propose a top-$K$ attribute prototype matching method. First, computing the similarity between the region feature $z_m$ and the category's multiple attribute prototypes as $\{\langle a_{c,1}, z_m \rangle, \langle a_{c,2}, z_m \rangle, \ldots, \langle a_{c,w}, z_m \rangle\}_{w=1}^{W}$. Then, selecting the top-$K$ similar attribute prototypes, assuming that these $K$ attribute prototypes can approximately represent the major attribute features of the region, and weighting these top-$K$ attribute prototypes based on their similarity as $\sum_{k=1}^{K} \langle a_{c,k}, z_m \rangle \cdot a_{c,k}$. Finally, the weighted similarities of each attribute prototype is

multiplied by the object prototype of the category $p_c$ to evaluate the importance of each attribute prototype in relation to the category's prototype representation. The final result is used as the attribute prototype matching similarity score. The similarity score between the $m$-th region feature $z_m$ and the attribute prototypes of category $c$ can be expressed as follows:

$$S_{\text{attr}}(p_c, a_c, z_m) = \left( \sum_{k=1}^{K} \langle a_{c,k}, z_m \rangle \cdot a_{c,k} \right) \cdot p_c \tag{6}$$

The final matching strategy for the $m$-th region feature $z_m$ in the image $I$, combining object prototype and attribute prototype matching, is as follows:

$$\hat{y}_m = \arg \max_{c \in C_{\text{test}}} \left( \langle w_c, z_m \rangle + \lambda_p S_{\text{prot}}(p_c, z_m) + \lambda_a S_{\text{attr}}(p_c, a_c, z_m) \right) \tag{7}$$

where $\lambda_p$ and $\lambda_a$ control the relative importance of the object prototype matching score and the attribute prototype matching score, respectively.

## 4 EXPERIMENTS

In this section, we briefly explain the experimental setup, including datasets and implementation details. Next, we evaluate the performance of HMKM compared with various models.

**Datasets.** We evaluated HMKM on two widely adopted datasets, i.e., COCO (Lin et al., 2014) and LVIS (Gupta et al., 2019). For the COCO dataset, we adopt the OVOD setting of OVR-CNN (Zareian et al., 2021), splitting the object categories into 48 base categories and 17 novel categories. It includes 118k images, with 107,761 designated for training and 4,836 for validation. Following VLDet (Lin et al., 2023), we report mean Average Precision (mAP) at an IoU of 0.5. For the LVIS dataset, following the OVOD setting of ViLD (Gu et al., 2022), we split the object categories into 866 base categories and 337 novel categories, and report the mask AP for all categories. For brevity, we denote the open-vocabulary benchmarks based on COCO and LVIS as OV-COCO and OV-LVIS.

**Implementation Details.** In our experiments, we employ models in recent studies as the baselines and integrate our HMKM method on them in a training-free manner for evaluations. For each OVOD model, we utilize its detector backbone to extract the corresponding object and attribute prototypes, ensuring the alignment between the feature space of knowledge and that of the model. The number of category images for generating object prototype knowledge $T$ is empirically set to 10, while $M$ for attribute prototype knowledge is empirically set to 50. Following MM-OVOD (Kaul et al., 2023), the primary source of category images is ImageNet-21k (Deng et al., 2009). If the number is insufficient, we randomly selecting additional images from the training sets of Visual Genome (Krishna et al., 2017) and LVIS. The random crop ratio employed for extracting attribute regions from category images ranges from 0.4 to 0.6. The clustering method used for category attribute prototype representation is K-means++ (Arthur & Vassilvitskii, 2006). The number of clusters $W$ in the production of attribute prototypes is set to 15. The $\lambda_p$ and $\lambda_a$ for object prototype matching and attribute prototype matching are set to 0.25 and 0.3 for OV-COCO, 0.2 and 0.05 for OV-LVIS, respectively. All expriments are conducted on 4 NVIDIA V100 GPUs. More details can be found in the Appendix.

### 4.1 BENCHMARK RESULTS

We evaluate the proposed HMKM on COCO and LVIS datasets in the OVOD setting and compare with various state-of-the-arts. The results are reported in Table 1 and Table 2.

**OV-COCO Benchmark.** As shown in Table 1, integrating HMKM can further improve the performance of various open-vocabulary detectors by incorporating hierarchical multimodal knowledge at both the object and attribute levels. For the Detic (Zhou et al., 2022), which uses weak supervision from image classification data to expand the detector's vocabulary, HMKM improves performance by 1.7AP$_{50}^{novel}$. For Codet (Ma et al., 2024a), which explores object co-occurrence to find region-word alignments in open-vocabulary detection, HMKM improves performance by 2.4AP$_{50}^{novel}$. For BARON (Wu et al., 2023a), which develops a neighborhood sampling strategy to group contextually

Table 1: Compared with existing OVOD models on COCO dataset with the RN50-C4 and RN50-FPN backbones. The $AP_{50}^{novel}$ is a primary indicator to reflect the performance. The best results are highlighted in bold.

| Method | Supervision | Backbone | $AP_{50}^{novel}$ | $AP_{50}^{base}$ | $AP_{50}^{all}$ |
|---|---|---|---|---|---|
| Detic (Zhou et al., 2022) | Image | RN50-C4 | 27.8 | 51.1 | 44.9 |
| + HMKM | Image | RN50-C4 | **29.5** | 50.8 | 45.3 |
| CoDet (Ma et al., 2024a) | Caption | RN50-C4 | 30.6 | 52.5 | 46.8 |
| + HMKM | Caption | RN50-C4 | **33.0** | 52.3 | 47.3 |
| VLDet (Lin et al., 2023) | Caption | RN50-C4 | 32.0 | 50.6 | 45.8 |
| + HMKM | Caption | RN50-C4 | **34.2** | 50.4 | 46.1 |
| BARON (Wu et al., 2023a) | CLIP | RN50-FPN | 34.0 | 60.4 | 53.5 |
| + HMKM | CLIP | RN50-FPN | **35.7** | 60.2 | 53.8 |

Table 2: Compared with existing OVOD models on LVIS dataset with the ResNet50 and Swin-B backbones. The $AP_{novel}^{m}$ is a primary indicator to reflect the performance. The best results are highlighted in bold.

| Method | Backbone | $AP_{novel}^{m}$ | $AP_{c}^{m}$ | $AP_{f}^{m}$ | $AP_{all}^{m}$ |
|---|---|---|---|---|---|
| Detic (Zhou et al., 2022) | RN50 | 21.3 | 30.9 | 35.5 | 31.0 |
| + HMKM | RN50 | **22.1** | 30.7 | 35.3 | 31.0 |
| VLDet (Lin et al., 2023) | RN50 | 21.7 | 29.8 | 34.3 | 30.1 |
| + HMKM | RN50 | **23.2** | 29.6 | 34.1 | 30.3 |
| CoDet (Ma et al., 2024a) | RN50 | 23.7 | 30.6 | 35.4 | 31.3 |
| + HMKM | RN50 | **24.3** | 30.3 | 35.1 | 31.1 |
| MM-OVOD (Kaul et al., 2023) | RN50 | 27.2 | 33.2 | 35.6 | 33.1 |
| + HMKM | RN50 | **28.0** | 33.1 | 35.4 | 33.1 |
| VLDet (Lin et al., 2023) | Swin-B | 26.3 | 39.4 | 41.9 | 38.1 |
| + HMKM | Swin-B | **29.2** | 39.1 | 41.7 | 38.4 |
| Detic (Zhou et al., 2022) | Swin-B | 33.8 | 41.3 | 42.9 | 40.7 |
| + HMKM | Swin-B | **35.4** | 41.0 | 42.7 | 40.7 |

related regions and uses contrastive learning to align these with pre-trained CLIP, HMKM improves performance by $1.7AP_{50}^{novel}$. After integrating HMKM, these OVOD models exhibit a slight performance decline on base categories due to overfitting during training. However, they still adequately recognize base categories while significantly enhance detection of novel categories, which is crucial in the OVOD setting and leads to an improvement in the overall $AP_{50}^{all}$. Our HMKM consistently enhances the performance across various training schemes and supervision types, demonstrating its general applicability to OVOD models.

**OV-LVIS Benchmark.** Table 2 presents performance comparisons on the LVIS dataset, demonstrating that our method improves performance in various cases. For models using ResNet50 as the backbone, our method consistently achieves an improvement of approximately $1.0AP_{novel}^{m}$. Specifically, MM-OVOD (Kaul et al., 2023), which builds multimodal classifiers using image exemplars and text descriptions, still gains an additional $0.8AP_{novel}^{m}$ with our HMKM. For models using Swin-B as the backbone, integrating our approach with VLDet (Lin et al., 2023) and Detic (Zhou et al., 2022) increases accuracy for novel categories by $2.9AP_{novel}^{m}$ and $1.6AP_{novel}^{m}$, respectively. The integration of HMKM into OVOD models significantly enhances the essential $AP_{novel}^{m}$, while largely preserving recognition performance for base categories. The results show that our method is able to improve the performance of existing state-of-the-art models across multiple datasets and various backbones, further demonstrating its effectiveness and generalizability.

## 4.2 ABLATION STUDY

**Effectiveness of Different Components.** We conduct ablation studies on OPM and APM, by integrating them into various baselines on the LVIS dataset to assess each matching's effectiveness. As shown in Table 3, both matching strategies consistently improve novel categories detection while

Table 3: Ablation study of HMKM on LVIS dataset.

| Method | OPM | APM | $AP^m_{novel}$ | $AP^m_c$ | $AP^m_f$ | $AP^m_{all}$ |
|---|---|---|---|---|---|---|
| VLDet (Lin et al., 2023) | | | 26.3 | 39.4 | 41.9 | 38.1 |
| | | ✓ | 27.2 | 39.3 | 41.8 | 38.2 |
| | ✓ | | 28.7 | 39.2 | 41.8 | 38.4 |
| | ✓ | ✓ | 29.2 | 39.1 | 41.7 | 38.4 |
| Detic (Zhou et al., 2022) | | | 33.8 | 41.3 | 42.9 | 40.7 |
| | | ✓ | 34.4 | 41.1 | 42.7 | 40.6 |
| | ✓ | | 35.1 | 41.1 | 42.8 | 40.7 |
| | ✓ | ✓ | 35.4 | 41.0 | 42.7 | 40.7 |

preserving accuracy for base categories. Specifically, OPM and APM increase the $AP^m_{novel}$ for VLDet (Lin et al., 2023) and Detic (Zhou et al., 2022) by 2.4/0.9 and 1.3/0.6, respectively. OPM enhances the detection of novel categories through object prototype matching to overcoming challenges in visual representation learning, while APM complements this by focusing on attribute prototype matching for novel categories, together improving the detection performance. Our method, HMKM, which integrates OPM and APM, is able to surpass the performance of either matching strategy used independently.

**Object Prototypes with Different Numbers of Images.** As shown in Figure 3, increasing the number of images for object prototype representation shows a similar trend across several models. With just a single image, there is a noticeable improvement, indicating that one-image object prototype is already effective. Using more than 5 or 10 images, the novel AP becomes saturated, and further increasing the number to 50 or 100 images does not lead to additional improvement. This suggests that 5 to 10 images are sufficient for the model to effectively recognize novel categories.

**Effectiveness of Attribute Clustering.** As shown in Table 4, using individual local regions as attribute prototypes even slightly reduces $0.1AP^{novel}_{50}$ compared with not using attribute prototype matching. However, employing cluster centers as attribute prototypes increases $0.7AP^{novel}_{50}$. This indicates that individual local regions as attribute prototypes might introduce noise, while cluster centers are more robust and representative.

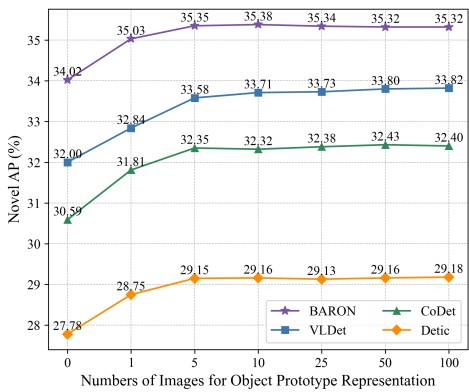

Figure 3: Comparing model performance on novel AP using object prototypes with different numbers of images.

**Top-$k$ in APM.** Top-$k$ is a key parameter in attribute prototype matching, determining how many of the most similar attribute prototypes are selected for attribute-level similarity measurement. Based on the CoDet model, we conduct experiments on the COCO dataset, as shown in Table 5. The experiments show that performance initially increases with $k$ and then decreases, reaching a peak at $k = 2$ for COCO, which we have adopted as the default setting.

Table 4: Effectiveness of Attribute Clustering.

| Strategy | $AP^{novel}_{50}$ | $AP^{base}_{50}$ | $AP^{all}_{50}$ |
|---|---|---|---|
| w/o | 32.3 | 52.5 | 47.2 |
| individual | 32.2 | 52.4 | 47.2 |
| cluster | **33.0** | 52.3 | 47.3 |

Table 5: Top-$k$ in APM.

| Top-$k$ | $AP^{novel}_{50}$ | $AP^{base}_{50}$ | $AP^{all}_{50}$ |
|---|---|---|---|
| w/o | 32.3 | 52.5 | 47.2 |
| 1 | 32.8 | 52.4 | 47.3 |
| 2 | **33.0** | 52.3 | 47.3 |
| 5 | 32.8 | 54.3 | 47.2 |

Table 6: Comparison with other methods using images in open-vocabulary detection under the same training-free setting.

| Method | $AP_{novel}^m$ | $AP_{all}^m$ |
|---|---|---|
| Baseline (Zhou et al., 2022) | 33.8 | 40.7 |
| MM-OVOD* (Kaul et al., 2023) | 34.0 | 40.7 |
| OVMR (Ma et al., 2024b) | 34.4 | 40.9 |
| HMKM | **35.4** | 40.7 |

Table 7: Comparison of mean inference time per image.

| Method | Backbone | Time (s) | $AP_{50}^{novel}$ |
|---|---|---|---|
| Detic (Zhou et al., 2022) | RN50-C4 | 0.1397 | 27.8 |
| + HMKM | RN50-C4 | 0.1415 | **29.5** |
| CoDet (Ma et al., 2024a) | RN50-C4 | 0.1429 | 30.6 |
| + HMKM | RN50-C4 | 0.1439 | **33.0** |
| VLDet (Lin et al., 2023) | RN50-C4 | 0.1432 | 32.0 |
| + HMKM | RN50-C4 | 0.1505 | **34.3** |
| BARON (Wu et al., 2023a) | RN50-FPN | 0.1084 | 34.0 |
| + HMKM | RN50-FPN | 0.1095 | **35.7** |

## 4.3 Further Analysis

**Training-Free Methods Comparison.** In the same training-free setting, we compare our HMKM with two image-based OVOD models, using the Swin-B version of Detic as a baseline. Notably, since MM-OVOD is not training-free, we use its classifier as an auxiliary, weighted to Detic's training classifier, similar to our HMKM. Results for OVMR are sourced from its original publication. Table 6 shows that HMKM is able to improve $AP_{novel}^m$ by matching representational knowledge from the model itself with region features, outperforming adaptations like MM-OVOD and OVMR that modify VLMs classifiers.

**Analysis of Inference Time.** In our setup, we analyze four OVOD models on the COCO dataset to determine the impact of integrating HMKM on single-image inference time. Table 7 shows that HMKM integration slightly increases the inference time by an average of 0.024s, yet it effectively improve the performance by 2.0$AP_{50}^{novel}$ when detecting novel categories.

Table 8: Transfer to other datasets. Evaluating COCO-trained model on the PASCAL VOC test set and LVIS validation set using mAP at IoU 0.5, without additional training.

Table 9: Analysis of multimodal knowledge independence on COCO dataset. HMKM-Base denotes that HMKM uses hierarchical multimodal knowledge from the base method.

| Method | PASCAL VOC | LVIS |
|---|---|---|
| Detic (Zhou et al., 2022) | 64.2 | 8.5 |
| + HMKM | 65.2 | 9.0 |
| CoDet (Ma et al., 2024a) | 65.4 | 11.1 |
| + HMKM | 66.7 | 11.6 |
| VLDet (Lin et al., 2023) | 65.3 | 11.5 |
| + HMKM | 66.1 | 12.2 |
| BARON (Wu et al., 2023a) | 65.9 | 12.1 |
| + HMKM | 66.8 | 12.7 |

| Method | $AP_{50}^{novel}$ | $AP_{50}^{all}$ |
|---|---|---|
| Base | 1.3 | 39.3 |
| + HMKM-Base | **3.9** | 40.0 |
| Detic (Zhou et al., 2022) | 27.8 | 44.9 |
| + HMKM-Base | **29.9** | 45.3 |
| CoDet (Ma et al., 2024a) | 30.6 | 46.8 |
| + HMKM-Base | **32.3** | 47.1 |
| VLDet (Lin et al., 2023) | 32.0 | 45.8 |
| + HMKM-Base | **32.8** | 64.2 |

**Transfer to Other Datasets.** To evaluate the generalization ability of our HMKM, we conduct experiments on transferring COCO-trained models to PASCAL VOC (Everingham et al., 2010) test set and LVIS validation set without additional training. We replace the class embeddings in the classifier head of the COCO-trained models with categories from these two datasets and integrate our HMKM for matching region features. PASCAL VOC includes 20 object categories, of which 9 are absent in COCO, complicating model transfer due to missing supplementary images and domain

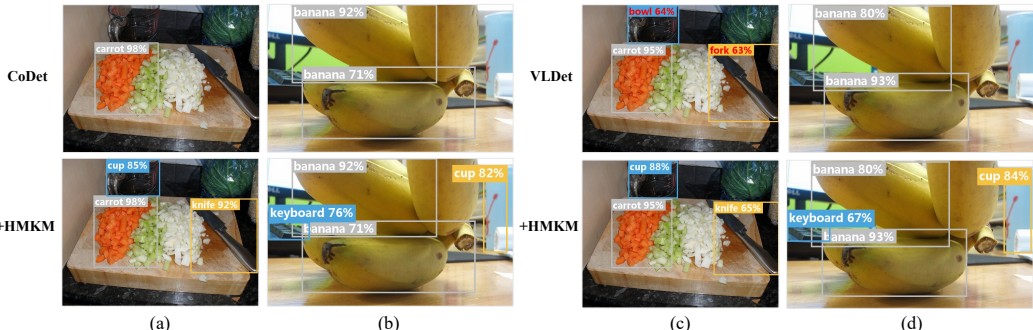

Figure 4: The visualization highlights HMKM's enhancement of novel categories detection on the COCO dataset compared to CoDet and VLDet. Grey boxes indicate base categories detections, while other colors denote novel categories. White text signifies correct detections, and red text indicates misclassifications.

gaps. Meanwhile, LVIS includes 1,203 categories, significantly expanding beyond COCO's label range. Experimental results in Table 8 show that integrating HMKM with existing models improves performance by $1.0AP_{50}^{m}$ on PASCAL VOC and $0.6AP_{50}^{m}$ in average on LVIS. These evidences demonstrate that the integrated HMKM effectively enhances the transfer learning performance of existing models without additional training.

**Analysis of Multimodal Knowledge Independence.** To validate multimodal knowledge independence, we employ a base method using Faster R-CNN (Ren et al., 2015), which is trained on the fully supervised detection data for COCO base categories with CLIP embeddings as the classifier head. We extract hierarchical multimodal knowledge from the base method and integrate it into multiple OVOD models using HMKM, denoted as the HMKM-Base method. As shown in Table 9, HMKM-Base not only improves the base model's detection of novel categories but also enhances multiple OVOD models in an unsupervised manner. For models like Detic and Codet, the improvement is comparable to or even exceeds that achieved using knowledge from their respective visual backbones. This indicates that hierarchical multimodal knowledge can be model-independent to some extent, further validating the effectiveness of our hierarchical multimodal knowledge representation and matching approach.

### 4.4 QUALITATIVE VISUALIZATION

Figure 4 shows that HMKM's hierarchical multimodal knowledge reasonably improves OVOD models' ability by accurately detecting challenging novel categories. With HMKM integrated, CoDet now can detect previously unnoticed objects, such as a cup and knife in scenario (a), and a keyboard and cup in scenario (b). Similarly, with HMKM integrated, VLDet can now correctly recognize a cup previously mistaken for a bowl and a knife previously mislabeled as a fork in scenario (c), and it can also detect a previously undetected keyboard and cup in scenario (d).

## 5 CONCLUSION AND FUTURE WORK

In this paper, we have presented HMKM, a training-free method inspired by the learning processes of human brains, leveraging hierarchical multimodal knowledge to represent novel categories and improve the capability of existing OVOD models when detecting novel categories. We have built object prototype knowledge for object-level matching, complementing the categories' textual descriptions. To better capture key visual features at the attribute level, we have also developed attribute prototype knowledge for fine-grained matching. Thus, our method integrates both object-level and attribute-level matching, significantly enhancing the performance of OVOD models without additional training, proving effective across various backbones and datasets. In the future, we aim to explore more effective knowledge representation strategies to reduce reliance on the quantity of images. Additionally, we plan to investigate adaptive methods for combining matching scores from object and attribute knowledge without adding extra hyperparameters.

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

# A APPENDIX

## A.1 ANALYSIS OF $\lambda_p$ AND $\lambda_a$ SELECTION

In this section, we analyze the selection of $\lambda_p$ and $\lambda_a$. Based on the ablation studies, object prototype matching demonstrates a marginal advantage over attribute prototype matching, although the two strategies share certain similarities. Consequently, in determining the values of $\lambda_p$ and $\lambda_a$, we initially set $\lambda_a$ to 0 and systematically adjust $\lambda_p$ to identify its optimal value. Once $\lambda_p$ is fixed at this optimal value, we subsequently adjust $\lambda_a$ until its optimal value is obtained. Based on Figure 5a, it can be observed that the impact of $\lambda_p$ on Novel AP follows a trend of initially increasing and then decreasing, with the maximum improvement occurring at $\lambda_p = 0.25$. By fixing $\lambda_p = 0.25$, Figure 5b further shows that $\lambda_a = 0.3$ yields the highest improvement in Novel AP. Thus, the optimal values for $\lambda_p$ and $\lambda_a$ are identified. The experiments are conducted with CoDet as the base model on the COCO dataset, and similar trends in hyperparameter influence have been observed across other OVOD models.

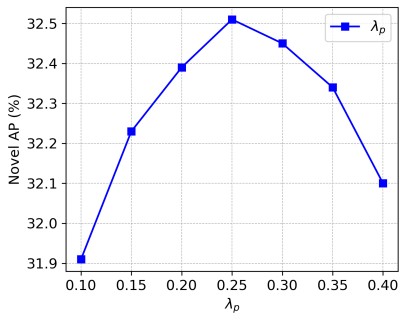 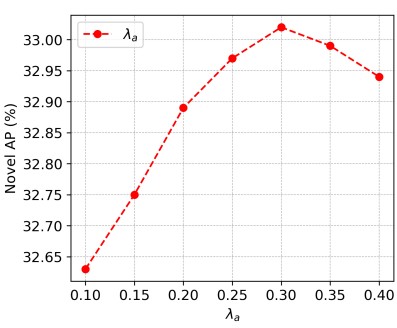

(a) $\lambda_p$ on the COCO dataset.      (b) $\lambda_a$ on the COCO dataset.

Figure 5: Analysis of $\lambda_p$ and $\lambda_a$ selection.

## A.2 ANALYSIS OF DIFFERENT ATTRIBUTE CLUSTERING METHODS

As shown in Table 10, the attribute prototype representation exhibits consistent performance across different clustering methods, showing relatively minor variations in effectiveness. This demonstrates that our attribute prototype representation is robust to some extent, as it does not rely on any specific clustering method. Therefore, for simplicity and consistency, we employ the commonly used KMeans++ as the default attribute clustering method throughout this paper.

Table 10: Analysis of different attribute clustering methods.

| Strategy | $\text{AP}_{50}^{novel}$ | $\text{AP}_{50}^{base}$ | $\text{AP}_{50}^{all}$ |
|---|---|---|---|
| w/o | 30.6 | 52.5 | 46.8 |
| GMM | 31.8 | 52.5 | 47.0 |
| KMeans | 31.9 | 52.5 | 47.1 |
| KMeans++ | 32.0 | 52.5 | 47.1 |
| Agglomerative | 32.1 | 52.5 | 47.1 |

## A.3 ANALYSIS OF DIFFERENT NUMBERS OF CLUSTERS

The data presented in Table 11 indicates a clear trend in $\text{AP}_{50}^{novel}$ as the number of clusters increases. Initially, from 5 to 15 clusters, there is a noticeable improvement in $\text{AP}_{50}^{novel}$, with values rising from 31.5 to 32.0. However, further increasing the number of clusters beyond 15 does not result in additional performance gains, as $\text{AP}_{50}^{novel}$ remains stable at 32.0. Given that the computational cost of clustering tends to increase with the number of clusters, it is both efficient and effective to select

15 clusters as the default in this analysis, balancing performance improvement with computational overhead.

Table 11: Analysis of different numbers of clusters.

| Number | $AP_{50}^{novel}$ | $AP_{50}^{base}$ | $AP_{50}^{all}$ |
|---|---|---|---|
| w/o | 30.6 | 52.5 | 46.8 |
| 5 | 31.5 | 52.5 | 47.0 |
| 10 | 31.8 | 52.5 | 47.1 |
| 15 | 32.0 | 52.5 | 47.1 |
| 20 | 32.0 | 54.5 | 47.1 |
| 25 | 32.0 | 52.5 | 47.1 |

### A.4 ATTRIBUTE PROTOTYPES WITH DIFFERENT NUMBERS OF IMAGES

As shown in Table 12, when the number of images increases from 10 to 50, the attribute prototypes result in an improvement in $AP_{50}^{novel}$. However, when the number of images increases from 50 to 100, there is no further noticeable improvement in $AP_{50}^{novel}$, and the performance even slightly decline, potentially due to the introduction of noise. Therefore, we use 50 images per category by default for constructing attribute prototypes.

Table 12: Attribute prototypes with different numbers of images.

| Number | $AP_{50}^{novel}$ | $AP_{50}^{base}$ | $AP_{50}^{all}$ |
|---|---|---|---|
| w/o | 30.6 | 52.5 | 46.8 |
| 10 | 31.7 | 52.5 | 47.0 |
| 50 | 32.1 | 52.5 | 47.1 |
| 100 | 31.9 | 52.5 | 47.1 |

### A.5 ANALYSIS OF THE CROP RATIO FOR ATTRIBUTE REGIONS

We observed that while attribute prototypes are generally robust to variations in the crop ratio for attribute regions, they are still affected to some extent. As shown in Table 13, when the crop ratio is between (0.2, 0.4), the improvement in $AP_{50}^{novel}$ is less than in the range of (0.4, 0.6), likely because the cropped regions are too fine-grained to be distinguishable. Conversely, when the crop ratio is between (0.6, 0.8), the attribute prototypes become more similar to object prototypes, failing to fully capture the fine-grained details of attribute matching, resulting in less improvement compared to the crop ratio range of (0.4, 0.6). Therefore, we adopt a crop ratio of (0.4, 0.6) as the default.

Table 13: Analysis of the crop ratio for attribute regions.

| Crop Ratio | $AP_{50}^{novel}$ | $AP_{50}^{base}$ | $AP_{50}^{all}$ |
|---|---|---|---|
| w/o | 32.5 | 52.5 | 47.2 |
| (0.2, 0.4) | 32.9 | 52.5 | 47.3 |
| (0.4, 0.6) | 33.1 | 52.5 | 47.3 |
| (0.6, 0.8) | 33.0 | 52.3 | 47.3 |

### A.6 VISUALIZATION OF CATEGORIES IMAGES

In this section, we present a subset of category images used to construct our hierarchical multimodal knowledge, as shown in Figure 6. It is evident that even within a single category, there are considerable variations, such as differences in lighting, changes in object orientation, and variations in background elements. However, our approach effectively represents the prototypes of categories in a hierarchical manner, which can then be integrated into the OVOD models to enhance performance.

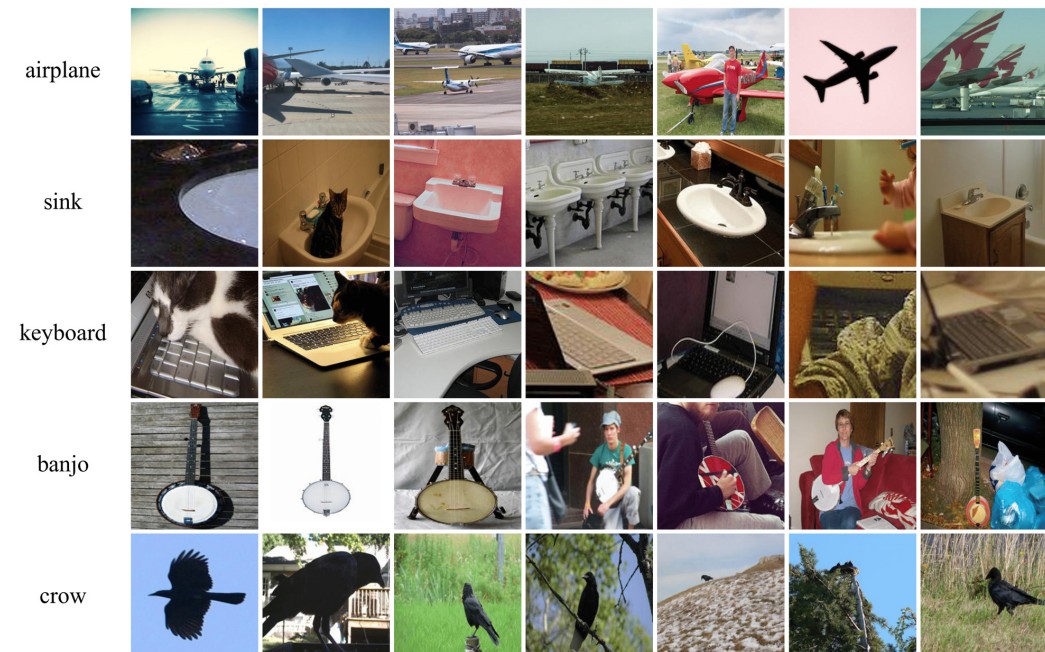

Figure 6: Visualization of categories images.

### A.7 FURTHER VISUALIZATION OF DETECTION COMPARISON RESULTS

In Figure 7, we visualize the improvements in novel categories detection after integrating HMKM. In each image set, the left column displays the result from recent OVOD models (e.g., CoDet, VLDet), while the right column shows the result after HMKM integration. Grey boxes indicate base categories detections, while other colors denote novel categories. White text signifies correct detections, and red text indicates misclassifications. Before HMKM integration, models miss many novel objects, such as an airplane, sink, and skateboard, and misclassify items like a keyboard as a bench. With HMKM integration, these detection issues are noticeably reduced.

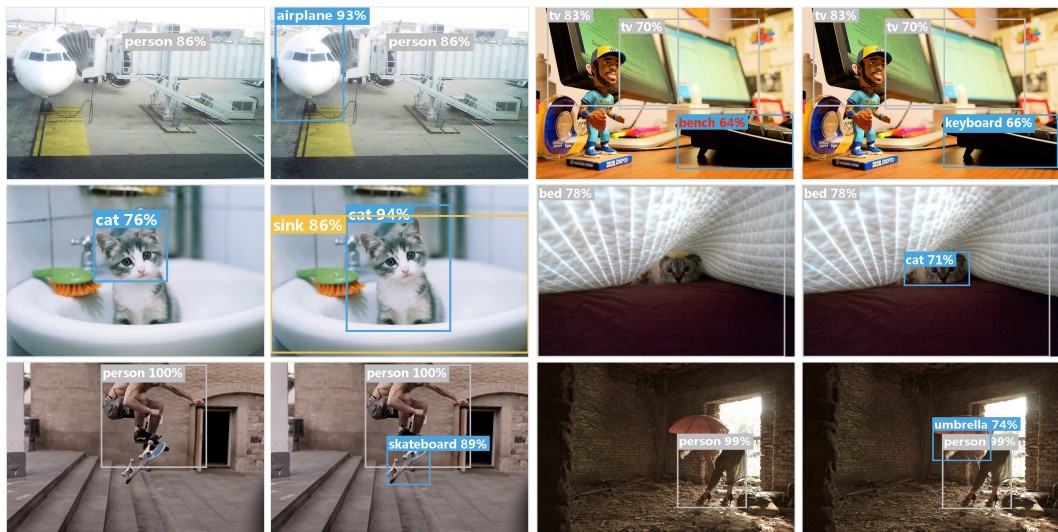

Figure 7: Further visualization of detection comparison results.

