# OpenReview forum: "Hierarchical Multimodal Knowledge Matching for Training-Free Open-Vocabulary Object Detection"
_ICLR.cc/2025/Conference — ICLR 2025 Conference Withdrawn Submission_

### Official Review · Reviewer_C39d · 2024-10-29

**Soundness:** 4
**Presentation:** 3
**Contribution:** 3
**Rating:** 5
**Confidence:** 4

**Summary:**

This work designs a prototype-based matching method, called HMKM, to help OVOD models distinguish novel categories. The proposed method includes a object branch and an attribute branch, which are adaptively combined. Experiments demonstrate that the proposed method improves the performance of various OVOD models on both COCO and LVIS datasets.

**Strengths:**

- The proposed method is straight-forward and can be applied on almost any existing OVOD models.
- Experiments are conducted on various OVOD, demonstrating the effectiveness of the method.
- Ablation studies are comprehensive and the authors have included visualizations to demonstrate the improvement in detection performance.

**Weaknesses:**

- Some parts in Sec. 3 are not clear enough. For instance, in eq. (3), what is the dimension of the output of phi_det? Since phi_det is the frozen OVOD backbone, I suppose its output should be (h, w, c). But as a prototype, it makes more sense if it is a vector instead of a 2D feature map. Is there any pooing or aggregation step to convert the feature map to a vector representation for the prototype?
- Line 306 states that, LVIS images may be used to compute prototypes. However, any information related to novel categories should have been deleted from the LVIS train split. In such case, how to find the LVIS images that contains novel categories? How are the prototypes computed without bounding box annotations of novel categories? It would be helpful if more details on the image selection process for novel categories are provided.

**Questions:**

From table 1, table 2, and other comparison results, I notice that the AP_all does not benefit from HMKM. Moreover, the performance of base categories seem to drop after HMKM is applied. However, I do not see any reason HMKM only works for novel categories. Also, in figure 4, the predictions for base categories are identical before and after adding HMKM. Does this suggest that HMKM is not applied to base categories? I wonder what if HMKM is also applied to base categories?

---

### Official Review · Reviewer_yFDB · 2024-10-30

**Soundness:** 3
**Presentation:** 3
**Contribution:** 2
**Rating:** 3
**Confidence:** 4

**Summary:**

This paper addresses open vocabulary object detection by utilizing category-specific images to create object and attribute prototype representations, which serve as a plug-and-play module for existing OVOD models, enabling category matching with region features without the need for extensive training.

**Strengths:**

1. The author aims to address OVD problem by training-free manner, which is interesting and useful for the OVD community.

2. The authors conduct several ablation experiments.

3. The paper is overall easy to read.

**Weaknesses:**

About the motivation:
1. It’s stated that training-based methods requires large-scale image-text datasets to alleviate unawareness for novel categories, and the proposed method aims to propose training-free manner to circumvent such weak supervision. However, the introduction of ImageNet-21k during inference phase impose a strong prior of having access to images repository (most existing OVD methods only assumes having access to the category names during inference), such relaxed setting and over-abundant prior information provided for the model might not be applicable in real-world OVD scenarios.

2. About the novelty: The proposed method uses the combination of attribute-level similarity and object-level similarity to get matching score. At high-level, using dual-path score during inference to modulate classification confidence score is not novel. For example, ViLD[1] and F-VLM [2] ensembles predictions of the detector and CLIP via the geometric mean.

[1] Gu, Xiuye, et al. "Open-vocabulary object detection via vision and language knowledge distillation." arXiv preprint arXiv:2104.13921 (2021).
[2] Kuo, Weicheng, et al. "F-vlm: Open-vocabulary object detection upon frozen vision and language models." arXiv preprint arXiv:2209.15639 (2022).

About the writing:
1. The proposed method refers the object-level and attribute-level features as hierarchical knowledge, while in essence, object-level and attribute-level features are extracted of different sizes of image crops, and whether they can be referred as “hierarchical” is dubious.
2. In broad context, compositional learning (CL) [3], which draws inspiration from the human capability to comprehend and create complex ideas from basic concepts, aims to endow machines with comparable abilities in understanding, reasoning, and learning, which is closely related to the focus of this paper. It has been successfully applied to various computer vision tasks including human-object interaction [4] and scene-graph generation [5], etc. The author did not include necessary summary including (How CL benefits task like object detection task in general? How ) in introduction part.

[3] “Compositional Learning: Perspectives, Methods, and Paths Forward” https://compositional-learning.github.io/
[4] Hou, Zhi, et al. "Detecting human-object interaction via fabricated compositional learning." Proceedings of the IEEE/CVF Conference on Computer Vision and Pattern Recognition. 2021.
[5] Li, Lin, et al. "Compositional feature augmentation for unbiased scene graph generation." Proceedings of the IEEE/CVF International Conference on Computer Vision. 2023.

3. Some concepts are confusing. in the paper, it’s mentioned paired object prototype knowledge (line 225) and attribute prototype knowledge triplets (line 260), while there is no further illustration of how to utilize such knowledge pairs and triplets. Will they participate in computing the corresponding representations? Or they just a “hierarchical” concepts?

Unsupported claims:
1.One motivation of this paper is to alleviate the performance gap between base categories and novel categories by getting rid of supervised-learning. (as stated in line 45 and figure 1a), however, the performance gap across all models are still large as shown in table 1.

**Questions:**

1. In both OPM and APM process, multiple images are required to obtain prototypical representations, so image collection are introduced in section 3.2. Does it mean the model needs to sample T (in equation (3)) and M (in equation (5)) images of all categories C from ImageNet-21k repository to obtain prototypical objects representations and prototypical attributes representations?
2. Generalization of the proposed methods. It’s mentioned that the method could be easily integrated into existing OVOD models, while most methods selected in main experiments are two-stage based detection models. Transformer-based OVD methods such as ovdetr[6] and cora[7] also shows competitive performance. Have you tried improvement with transformer-based detectors?

[6] Zang, Yuhang, et al. "Open-vocabulary detr with conditional matching." European Conference on Computer Vision. Cham: Springer Nature Switzerland, 2022.

[7] Wu, Xiaoshi, et al. "Cora: Adapting clip for open-vocabulary detection with region prompting and anchor pre-matching." Proceedings of the IEEE/CVF conference on computer vision and pattern recognition. 2023.

---

### Official Review · Reviewer_tN1W · 2024-10-31

**Soundness:** 3
**Presentation:** 2
**Contribution:** 1
**Rating:** 5
**Confidence:** 4

**Summary:**

This paper proposes a method called HMKM to match image region features with categories. HMKM represents categories at two granularities and can be easily integrated as a plug-and-play module into existing models for Open-Vocabulary Object Detection to improve the detection performance of novel categories.

**Strengths:**

1)	The method forms effective representations of categories at both the object level and attribute level, which is beneficial for matching features with categories and mitigating confusion when dealing with similar objects.
2)	The method improves the performance of the OVOD model without requiring additional training or incurring significant computational overhead during inference.
3)	The paper is well-written with clear language and has very few typos.

**Weaknesses:**

1)	The innovative contribution relative to existing studies appears limited. While this work has improved the performance of the existing OVOD architecture, the novelty of this study is somewhat difficult to identify. Numerous related works[1] already utilize visual prototype knowledge to enhance model classification ability. A more explicit discussion on how it differs significantly from related methods should be conducted.
2)	The paper lacks an in-depth analysis of the category attribute prototype. Although the method introduces the category attribute prototype to reduce confusion between similar categories, there is insufficient analysis on how attribute prototype plays a role in identifying objects in similar but distinct categories, such as quantitative assessing the difference between objects in similar categories in attribute prototype matching or a case study on a pair of visually similar categories, showing how the attribute prototypes differ and impact classification.
3)	The declined performance on base categories need more detailed analysis. The authors attributes this phenomenon to overfitting, while the reason why overfitting has a more negative impact on the base categories’ performance than on the novel categories’ which are included in the training set needs a deeper explanation. Please explain why this phenomenon occurs by comparing the feature distributions of the base vs. novel categories before and after applying HMKM, or through other reasonable means.
4)	There are ambiguities in the equations. In Eq (5), $\text{Crop}(I_{m,n})$ seems to indicate the region obtained by cropping the image numbered (m, n), while the actual meaning should be the region obtained from the n-th cropping of the m-th image. Please reformulate the equation to clarify the intended meaning.
5)	More experiments can be conducted to demonstrate the effectiveness of deploying the method in more and closer OVOD models, such as: DetCLIPv3[2], YOLO-World[3], SHiNe[4] and so on.

[1] Yang Y, Ma C, Ju C, et al. Multi-modal Prototypes for Open-World Semantic Segmentation[J]. International Journal of Computer Vision, 2024: 1-17.
[2] Yao L, Pi R, Han J, et al. DetCLIPv3: Towards Versatile Generative Open-vocabulary Object Detection[C]//Proceedings of the IEEE/CVF Conference on Computer Vision and Pattern Recognition. 2024: 27391-27401.
[3] Cheng T, Song L, Ge Y, et al. Yolo-world: Real-time open-vocabulary object detection[C]//Proceedings of the IEEE/CVF Conference on Computer Vision and Pattern Recognition. 2024: 16901-16911.
[4] Liu M, Hayes T L, Ricci E, et al. SHiNe: Semantic Hierarchy Nexus for Open-vocabulary Object Detection[C]//Proceedings of the IEEE/CVF Conference on Computer Vision and Pattern Recognition. 2024: 16634-16644.

**Questions:**

See Weaknesses.

---

### Official Review · Reviewer_r8sj · 2024-11-02

**Soundness:** 3
**Presentation:** 3
**Contribution:** 3
**Rating:** 6
**Confidence:** 5

**Summary:**

The paper introduces a novel method, HMKM, for open-vocabulary object detection (OVOD) that leverages pre-trained vision-language models. HMKM aims to enhance the detection of novel categories without additional training. The method includes two main components: object prototype knowledge for object-level matching and attribute prototype knowledge for fine-grained, attribute-level matching. The paper demonstrates that HMKM, as a plug-and-play module, significantly improves the performance of various OVOD models across different datasets and backbones.

**Strengths:**

1. The paper presents a creative approach to OVOD by introducing a hierarchical multimodal knowledge matching method that combines object and attribute-level knowledge, which is a novel contribution to the field.
2. HMKM addresses a critical challenge in OVOD by improving the detection of novel categories without additional training, which is significant for practical applications where labeled data for new categories may be scarce.

**Weaknesses:**

1. The paper could further discuss the scalability of the HMKM approach, especially as the number of novel categories increases. While the method shows promising results, it is unclear how it would perform with a significantly larger vocabulary.
2. The paper could benefit from a deeper analysis of the computational overhead introduced by HMKM, particularly in terms of inference time and memory usage, which are critical for real-time applications.

**Questions:**

1. How does HMKM scale with the number of novel categories, and does it maintain performance as the vocabulary size increases significantly?
2. Could the authors provide more details on the computational overhead introduced by HMKM in terms of inference time and memory usage?
3. Is there potential to integrate HMKM with other modalities, such as depth or video data, to further improve OVOD performance?
4. The paper mentions the independence of multimodal knowledge to some extent. Can HMKM transfer knowledge between domains effectively, and if so, how does this impact performance?

---

### Official Review · Reviewer_TDF4 · 2024-11-05

**Soundness:** 4
**Presentation:** 2
**Contribution:** 2
**Rating:** 6
**Confidence:** 4

**Summary:**

This paper presents Hierarchical Multimodal Knowledge Matching (HMKM), a training-free approach that enhances Open-Vocabulary Object Detection (OVOD) models in recognizing novel categories. HMKM leverages hierarchical multimodal knowledge, integrating object and attribute prototype matching for improved detection accuracy. Experiments show that HMKM boosts performance across datasets, improving AP50 by 1.0 on PASCAL VOC and by 0.6 on LVIS, and it works effectively across various model backbones without additional training. HMKM’s model-independent framework offers a substantial improvement in detecting previously unseen categories.

**Strengths:**

1. The HMKM method is innovative in its training-free, hierarchical knowledge matching approach for open-vocabulary object detection. By combining object-level and attribute-level prototypes, it addresses a critical gap in novel category detection effectively.

2. The proposed method is rigorously tested on diverse datasets, including COCO, LVIS, and PASCAL VOC. Across these datasets, HMKM consistently improves detection performance, demonstrating the robustness and effectiveness of its design.

3.  The paper is well-organized, offering clear explanations of the hierarchical matching approach (with object prototype and attribute prototype modules) and visually demonstrating improvements in novel category recognition.

4. HMKM’s plug-and-play feature makes it easy to integrate into existing models, providing a direct way to enhance open-vocabulary detection without requiring additional training, which can be valuable for future applications in practical settings.

**Weaknesses:**

1. HMKM shows improvements on COCO and LVIS datasets, but it lacks comparisons with a wider range of state-of-the-art OVOD methods. Adding results from more recent OVOD models and using diverse datasets would better demonstrate HMKM’s effectiveness and generalizability.

2. The impact of object-level and attribute-level prototype matching is not thoroughly examined. Ablation studies on each component and different matching configurations would clarify their contributions to HMKM’s performance.

3. Although training-free, HMKM’s complex matching process may hinder computational efficiency. Discussing its scalability, memory usage, and inference time compared to traditional methods would clarify its suitability for real-world applications.

4. The dependence on CLIP embeddings could limit HMKM’s flexibility in domains where CLIP is less effective. Exploring alternatives or adaptive integration of domain-specific embeddings would make HMKM more versatile.

5. The qualitative analysis of HMKM’s impact on novel category detection could be more detailed. Highlighting specific cases of fine-grained errors and areas where HMKM struggles would provide clearer insights into its limitations.

6. While HMKM claims multimodal knowledge independence, this is not rigorously analyzed. Examining the interactions between CLIP embeddings and detection models would clarify HMKM’s generalizability across models and datasets.

**Questions:**

1. Since HMKM depends on CLIP for embedding representations, have the authors considered alternative embeddings or methods to adapt to domains where CLIP might be less effective?

2. The paper’s qualitative analysis could benefit from a breakdown of errors in specific cases. Could the authors provide more details on fine-grained misclassifications and scenarios where HMKM underperforms?

3. The authors claim multimodal knowledge independence, but the evaluation lacks rigorous support. Could they provide additional insights or theoretical analysis into how HMKM achieves this independence across models and datasets?

4. Could the authors expand the evaluation to include more recent OVOD models and additional datasets with unique object categories?

---

### Note · Authors · 2024-11-15

I have read and agree with the venue's withdrawal policy on behalf of myself and my co-authors.